# The Interface between the State and NGOs in Delivering Health Services in Zimbabwe—A Case of the MSF ART Programme

**DOI:** 10.3390/ijerph20237137

**Published:** 2023-12-03

**Authors:** Blessing Magocha, Mokgadi Molope, Martin Palamuleni, Munyaradzi Saruchera

**Affiliations:** 1Population and Health Research Entity, North-West University, Mafikeng 2735, South Africa; mokgadi.molope@nwu.ac.za (M.M.); martin.palamuleni@nwu.ac.za (M.P.); 2Africa Centre for HIV/Aids Management, Stellenbosch University, Stellenbosch 7602, South Africa; msaruchera@sun.ac.za

**Keywords:** antiretroviral therapy, health services, interface, state, NGOs, partnership, sustainability

## Abstract

An over-reliance on donor funding for HIV/AIDS healthcare services remains a concern in Africa. This study, therefore, explores the partnership between the Zimbabwean government and an international non-governmental organisation in delivering HIV/AIDS healthcare services. An interpretivist paradigm and descriptive phenomenological design were used to elicit the opinions, perceptions, and experiences of forty purposively sampled key informants. Thematic analysis was employed using ATLAS.ti version 7.1.4 to analyse the data. The differences in terms of policies, structures, and administrative issues between the partners identified challenges in the implementation of the programme. This was demonstrated through the reversal of the gains attained in prevention, care, and treatment. This raises concerns for increased risk of defaulters, drug resistance, and deaths. Therefore, the partners in this endeavour should negotiate an aligned approach for the efficient delivery of HIV/AIDS healthcare services.

## 1. Introduction

In the realm of global health, the interface between governments and non-governmental organisations (NGOs) has emerged as a dynamic arena of collaboration and contention. NGOs have become major role players alongside governments, especially in low- and middle-income countries, in fulfilling the social contract between people and their governments [1,2]. This intricate partnership, often seen as pivotal to healthcare delivery in resource-constrained settings [1,3,4], holds the potential to address critical health challenges, where healthcare systems grapple with multifaceted challenges [5]. One notable player in this field, Médecins Sans Frontières (MSF), has earned international acclaim for its commitment to providing vital healthcare services to vulnerable populations across the globe [6].

Like many African nations, Zimbabwe faces a complex array of health challenges [3,4]. These challenges are exacerbated by economic instability, political turbulence, and a high prevalence of HIV/AIDS [7]. In response to these pressing concerns, MSF, an international medical humanitarian organisation, initiated the provision of antiretroviral therapy (ART) in Zimbabwe in 2000. This programme aimed to combat the HIV/AIDS epidemic and improve overall health outcomes of the Zimbabwean population.

This study is underpinned by the Principal–Agent Theory and delves into the nuanced dynamics that govern the interface between the Zimbabwean state and MSF in the delivery of healthcare services, with a specific focus on the ART programme. Zimbabwe experiences many challenges, such as weak coordination mechanisms in terms of aid, donor dependency, and lagging in providing universal access to treatment, care, and support for HIV/AIDS patients. The collaboration between the state and NGOs in healthcare delivery is often two-faced, presenting opportunities for innovation and progress but also raising questions about accountability, sustainability, and the allocation of resources [1]. Within this context, the MSF ART programme serves as a compelling case study through which to explore the broader issues surrounding the partnership between governments and NGOs in healthcare.

This study, therefore, explores the partnership between the Zimbabwean government and MSF in delivering HIV/AIDS healthcare services. The study taps into the experiences of the participants to explore the partnership dynamics, impact, and outcomes of the MFS ART programme. It addresses two questions: How do the Zimbabwean government and MSF navigate the complexities of their partnership in delivering HIV/AIDS healthcare services, particularly ART? How has the MSF ART programme exerted an impact on HIV/AIDS healthcare in Zimbabwe?

By examining these questions, this study contributes to an informed understanding of the intricate interplay between state institutions and NGOs in the realm of healthcare delivery in Zimbabwe. Moreover, it proffers valuable insights that could inform not only the ongoing efforts to combat HIV/AIDS in Zimbabwe but also the broader contentions surrounding global health partnerships. In doing so, this study contributes to the ongoing dialogue on effective and sustainable approaches to HIV/AIDS healthcare delivery in resource-constrained settings worldwide.

### 1.1. Background

#### 1.1.1. Global Overview of Aid in Healthcare Systems

Over the past two decades, the effectiveness of donor aid has been at the centre of the development discourse. Amongst others, the following global summits and conferences are significant for their role in interrogating the efficacy of international aid as an instrument for human development: the Millennium Summit in 2000, where heads of state at the United Nations (UN) General Assembly adopted the Millennium Declaration as well as accepted the Millennium Development Goals (MDGs) as a monitoring framework for progress on key development indicators; the UN conference ‘Financing for Development’ in 2002; the Monterrey Consensus (March 2002), which called on developed countries to provide more and better aid, as well as improved trade and debt policies [8]; the Paris Declaration on aid effectiveness in 2005 [9]; and the fourth high-level forum on aid effectiveness in Busan in 2011 [10]. Thematically, these global summits and conferences highlighted the effectiveness of aid as it is affected by ownership, alignment, and harmonisation of management for positive results [8]. In addition, a vast body of empirical literature has been written about the impact of aid on development [11,12]. Although there is a growing interest from the global community to make aid effective, there is still no consensus on its positive impact and effectiveness on the development process [13,14]. On the other hand, some economists and development experts argue that aid has an essential role in promoting sustainable development [9].

Globally, aid has influenced development, particularly in impoverished nations. It has been instrumental in human development across the world. The areas of development where aid has had a significant impact include social transformation, infrastructural development, technology and management expertise, and creation of global partnerships and organisational forms to share and develop knowledge [10]. In most cases, aid has been used to build post-conflict societies, to alleviate humanitarian emergencies, and to support the strategic and commercial interests of the donor agencies [15,16]. Aid has thus contributed towards socio-economic development in developing countries [17].

As a result of donor aid, there has been success with major epidemics such as river blindness, smallpox, polio, and malaria [16]. This success could be attributed to partnerships between African governments and foreign donors [18]. Concerted efforts by the World Health Organisation (WHO) led to massive eradication of smallpox, which had claimed millions of lives more severely in impoverished regions in Africa. In absolute terms, smallpox struck the lives of 10 to 15 million people each year globally, and claimed 1.5 to 2 million lives [16]. These high figures caught the attention of the WHO, which established the Smallpox Eradication Unit (SEU), mandated with vaccinating the world population against smallpox. Another example in world health where aid played a pivotal role is the Global Alliance for Vaccines and Immunisation (GAVI). Donors were instrumental in introducing immunisation training and facilities in poor countries that could not afford the costs of vaccines and immunisation. Notable donors were the Bill and Melinda Gates Foundation (BMGF) in the 1990s, which gave an initial sum of USD 750 million, and The Global Alliance for Vaccines and Immunisations, founded in 2000 [16]. These donors have made inroads insofar as children’s vaccination is concerned. Using sophisticated technologies and systems of mass distribution, the donors, as of 2004, helped vaccinate 41.6 million children against hepatitis B; 5.6 million children against Haemophilus Influenza Type B; 3.2 million children against yellow fever; and 9.6 million children with other essential vaccines [16].

#### 1.1.2. Aid in African Healthcare Systems

In Africa, aid has become a cog in driving development. As [18] argues, aid is an integral part of Africa’s past and an essential part of its future. African governments benefit from aid as they get resources for making multiple investments in health, education, and economic infrastructure needed to break the vicious cycles of deprivation, dependency, and poverty [19]. Hence, in some African countries, aid has improved public management and service delivery, all closely associated with poverty reduction, improved social services, and competent public institutions [19].

Aid has remained a key source of external finance in the least developed states, ahead of remittances [20]. The least developed countries, primarily found in Africa, heavily depend on aid because they suffer from significant resource constraints. It is also difficult for them to access other external resources such as foreign direct investment, hence donor aid remains a vital source for funding development [21]. It is encouraging that aid to fragile states has become less volatile and more predictable over the past decade [20]. Recently, there was a significant increase in net Official Development Assistance (ODA) flows to Africa from USD 20.4 billion in 2002 to a peak USD 50.7 billion in 2011 and USD 46.1 billion in 2012 [22]. There is evidence that in the last three decades sub-Saharan Africa has accounted for a large proportion of the ODA disbursed. Africa alone received more than 30% of the total ODA [21]. According to UNCTAD [22], African countries will continue to depend on aid in the short and medium term. In addition, humanitarian assistance to Africa has been rising each year since 1998, resulting in nine countries on the continent positioned in the top fifteen list of highest recipients [23].

Aid in sub-Saharan Africa has been helpful in addressing several challenges bedevilling healthcare delivery [24]. African health systems rely heavily on donor funding [25,26]. The allocable donor assistance for health (DAH) increased steadily in 2010, and almost USD 8.1 billion has been invested in the health systems by donors [26]. African countries are thus heavily dependent on donor support to strengthen the health sector. Evidence shows that governmental healthcare expenditure in sub-Saharan Africa is grossly behind the target set by the World Health Organisation (WHO) [24]. Most countries lag behind in the fulfilment of the Abuja Declaration, where African countries agreed to commit 15% of their national budgets to health. Only six countries in the African region spend at least 15% of their national budget on health, namely Rwanda, 18.8%; Botswana, 17.8%; Niger, 17.8%; Malawi, 17.1%; Zambia, 16.4%; and Burkina Faso, 15.8% [26]. Furthermore, thirty-two out of fifty-three African Union (AU) member states spend less than the USD 40 recommended per person by the WHO, with eleven investing just USD 5 or less per capita [27].

It Is evident from the above literature that African countries need to commit to their healthcare obligations. The proliferation of donors is inevitable as they fill the health funding gap that African states have disowned. For example, in the 1980s, the Kenyan Government admitted that NGOs were responsible for providing 40% of all health facilities [28]. By extension, aid to healthcare from the OECD has risen, particularly to respond to the AIDS crisis and the emphasis in the MDGs on reducing the incidence of preventable diseases as well as maternal and child mortality [29].

#### 1.1.3. Aid in Zimbabwean Healthcare Systems

Zimbabwe receives foreign aid from many countries and international monetary agencies. It was a preferred destination for aid [30] until the political turmoil of the early 2000s led to donors withholding and even withdrawing support. The consequences have been less aid disbursed to Harare. In the first decade of independence, Zimbabwe was preferred by donors, the darling of Western donors who were prepared to pour about USD 300 million into the country each year [30]. Most of the celebrated cases of health and education delivery in the first decade of independence were achieved in the context of these partnerships between donors and the government in socio-economic development programmes [31,32].

However, during the 1990s, relations between the government and donors soured due to alleged human and property rights violations. Coupled with mistrust and the political turmoil of the early 2000s in the country, donors withheld aid, resulting in less aid disbursed to Harare [31,32,33]. Despite the tense and strained donor–government relations, aid remained a vital component to socio-economic development in Zimbabwe. Several aid-funded programmes and projects supporting maternal and child health, access to and quality of water supply, and sanitation to mitigate the impact of epidemics like HIV have been implemented [19,34]. For instance, health aid increased to 49% from 2002 to 2004 and the Ministry of Health confirms that donors were a significant source of funding for health [35].

Despite the much-publicised human and property rights violations, in 2000, donors such as DANIDA, NORAD, and the European Union disbursed funds to sustain the health budget through the Health Services Fund [31]. The health sector continues to rely heavily on donor support; many drugs come as donations and are thus procured externally [36]. The Health Transition Fund pays for the purchase of 98% of the drugs, while the remaining 2% are paid for by the AIDS Levy managed by the National AIDS Council [34]. In Zimbabwe, donor driven projects have helped vulnerable groups such as orphans affected by hyper-endemic HIV [19]. In addition, Zimbabwe has benefited from technical and capacity building projects funded by different donors [37], which has seen many projects undertaken in agriculture, power stations, and railway lines [38]. Often, the government of Zimbabwe has no capacity to run a complex set of programmes, so donors, on many occasions, have become providers [39]. For instance, donor funding has improved availability of medicines and medical staff in healthcare facilities across Zimbabwe [34].

The Government of Zimbabwe (GoZ) put structures to ensure that aid is effectively and efficiently managed. The main system is the National AIDS Council (NAC) and the National AIDS Trust Fund, popularly known as the AIDS Levy. The NAC is responsible for coordinating and implementing programmes and measures to combat HIV/AIDS. On the other hand, the AIDS Levy provides financial support to key HIV/AIDS interventions in Zimbabwe as well as complementing the external funding of HIV/AIDS activities in Zimbabwe [40]. In addition, the Aid Coordination Policy (ACP) was enacted in May 2009 to provide a framework for enhanced aid effectiveness and accountability [36]. It was designed in line with the principles of the Paris Declaration on Aid Effectiveness and the Accra Agenda for Action [36]. Furthermore, it was created to minimise duplication and align aid with national development plans and priorities by building institutional frameworks designed to improve the effectiveness of aid in Zimbabwe [41]. The other equally important objective was to re-orient aid from humanitarian to development assistance [41]. The Government Development Forum (GDF) was also created to promote dialogue between governments and donor partners.

However, the Aid Coordination Policy suffered major setbacks, including lack of capacity, fights for control by multi parties and line ministries during the span of the GNU, and reluctance by some donor countries to engage directly with the government, rendering it ineffective [39]. The government of Zimbabwe has continued to be vulnerable with no positive balance of payment support from major multilateral and bilateral institutions or donors due to substantial debt arrears of over USD 7 billion and an almost USD 2 billion domestic debt [42]. Other sectors have been severely affected as well, especially the health sector which has been in decline, resulting in a decrease in most basic services and a rising maternal and child mortality rate [35,43].

### 1.2. Statement of the Problem

The Zimbabwe government is expected to provide a policy framework in partnership with NGOs for healthcare delivery. This means that if this partnership is duly enforced, aid could achieve its intended aim in Zimbabwe. Aid has not met its expected goals in terms of healthcare delivery due to political and economic challenges. The gap in aid administration has had severe consequences, such as the intensification of diseases, untimely deaths, and unequal access to healthcare services. The failure to meet their expected goals has further aggravated more ravaging healthcare problems. That is, despite government and NGOs interventions, the problem persists. This brings to the fore the relevance of this study, which examines how the interface between the state and NGOs could curb the inherent problems in delivering health services in Zimbabwe. Research in this direction contributes to knowledge on how mutual collaborations between the state and NGOs could salvage most of the health challenges confronting people in Zimbabwe. This is because most studies that have examined this similar conundrum have merely focused on the role of the state and NGOs on an individual level. Consequently, there is little evidence and scant research on how a mutual connection between the state and NGOs could enhance the distribution of aid to resolve healthcare challenges in Zimbabwe. It is therefore worrisome that extant studies that have investigated this subject hardly interrogated why healthcare aid fails to achieve positive healthcare outcomes in Zimbabwe.

### 1.3. Theory

This study is underpinned by the Principal–Agent Theory, which provides a valuable lens to understand the complex dynamics between the Zimbabwean government and MSF in delivering healthcare services, specifically through the ART programme. Through this theory, this study explores how the delegation of healthcare service delivery to NGOs such as MSF exerts an impact on the government’s ability to achieve its healthcare goals and how this partnership addresses issues of accountability, information sharing, and interests. It is the duty of the government to ensure a well-structured and coordinated involvement of non-state actors to avoid the ‘dangers of special interest politics’ [1]. This study analyses the interests of both the principal and the agent (the government of Zimbabwe, represented by the Ministry of Health and Child Care (MoHCC), and MSF, respectively). This analytical framework provides insight into the complexities of state–NGO collaboration in healthcare delivery. It anchors the possibilities of perfecting such partnerships for better health outcomes in Zimbabwe and similar contexts.

## 2. Materials and Methods

This study adopted a qualitative research approach, producing descriptive data derived from participants’ own words [44]. The interpretive paradigm was used to define and redefine the meanings that the interviewer observed and heard from the participants of this study [45]. The descriptive phenomenological design was used to explore the challenges and risks in the implementation of the MSF ART programme in Zimbabwe, and how they can be addressed through the research participants’ opinions, perceptions, and experiences [46]. Overall, we elicited in-depth data to attain the research objectives.

### 2.1. Setting and Overview of the MSF ART Programme Transition to MoHCC

This study used as a case study the MSF programme on ART, which started in 2000, with activities implemented in the eight districts of Buhera, Epworth, Gutu, Beitbridge, Tsholotsho, Chikomba, Bulawayo, and Gweru in Zimbabwe. MSF started working in Zimbabwe in 2000 in partnership with the Ministry of Health and Child Care (MoHCC) at central, provincial, and district levels to provide access to ART for the first time to HIV/AIDS patients in Zimbabwe. According to the MSF Report of July 2012 [6], at the end of 2011, MSF supported about 48,430 clients on antiretroviral (ARV) treatment. Buhera had the highest number of beneficiaries, with 18,590. In Epworth, 14,220 beneficiaries were under treatment, while in Bulawayo and Gweru, 11,000 patients participated in the programme in each region. In Tsholotsho, 9000 patients were on ARV treatment. The MSF project in Beitbridge had 2500 patients. All the beneficiaries were fully integrated into the National Health Systems (NHS) in 2012 when the MSF programme was handed over to the government. The MSF ART programme was selected for this study because it transitioned from donor funding to government funding under the MoHCC. At the time of data collection, MSF had given the government autonomy to fund and manage the ART programme.

### 2.2. Sampling Technique

Two districts (Buhera and Tsholotsho) were selected by random sampling from the six districts in which the MSF project was implemented. These two districts are predominantly rural, and both have very high HIV prevalence compared to other districts in their respective provinces [47]. The people barely meet their nutritional requirements due to irregular food supplies [47]. In addition, the healthcare systems in both districts have a long history of getting funding from donors [48]. Secondly, purposive sampling was used to recruit participants for this study. Participants who were deemed likely to offer valuable insights into the study were selected [49]. In brief, purposive sampling occurred at more than one level. First, purposive sampling was employed to choose key informants. The key informants were those considered to have more direct, personal knowledge of the MSF ART programme and government policies, also referred to as data-rich sources [50]. Secondly, the research questions and objectives were used to purposively identify and determine the type of participants recruited for in-depth interviews. The research questions required someone with experience in the programme, and this is how the key informants were identified, based on their knowledge and experience. In brief, this study included only people who had direct knowledge of the MSF programme, either by being beneficiaries of MSF (workers with more than five years’ experience and ART patients), or working for the Ministry of Health (nurses and medical doctors), local government (district administrators), or other organisations which operated in the same districts with MSF.

In this study, the number of participants was determined by saturation, which is reached when gathering fresh data no longer brings new insights [49,51]. For this study to reach the saturation point, 40 participants were interviewed. These included two district administration officers (DAs), two district AIDS Committee (DAC) members, two district medical officers (DMOs), two district nursing officers (DNOs), four project managers of different non-government organisations (NGOs), fifteen nurses, and six MSF staff. Furthermore, seven informants were chosen from the provincial and national MoHCC offices. The different categories of informants were selected to obtain divergent views. Informed consent was sought from all sampled participants.

### 2.3. Data Collection

A survey was used to collect data comprising open-ended questions. The survey elicited information from key informants involved in ART activities in both districts. The questions were administered through in-depth interviews conducted with key informants. The questions were designed to elicit broad and general responses from the participants. Open-ended questions allowed for further probing and eliciting more information. The intention was to generate data in the participants’ own words on how they interpreted and understood the MSF ART programme’s transitional challenges and exposure to risks. Individuals’ responses to questions were also used to corroborate the information generated during the document review. The individual responses were recorded and transcribed.

The aim of the interviews was to elicit as much narrative as possible about the participants’ perceptions, knowledge, and beliefs about the MSF ART programme’s challenges and exposure to risks. In other words, this study sought their perception of the programme’s sustainability after the withdrawal of donor funding, providing insight into the participants’ views on the present and future of the MSF ART programme. They were asked questions on the resources brought into the programme by donor aid and how they would manage once donor funding was withdrawn. Although interviews did not record the participants’ life history, increased attention was paid to past experiences and changes in the programme as fieldwork proceeded. Emphasis was put on the knowledge they acquired, their experiences, and how their understanding developed throughout the life cycle of the programme from 2004 to 2015.

### 2.4. Data Analysis

A thematic approach was used to analyse the data. Transcripts were analysed by carefully reading and re-reading the data to identify emerging themes. Quotations of those themes were extracted from the recorded text [52]. The results of this study were analysed as an ongoing process while more data were being acquired. The first step of analysis took place after the interviews with participants, when the interviewers decided on areas of interest which needed further probing. This enabled the researchers to constantly review the data. Lastly, transcripts and notes were reviewed to generate critical ideas and major categories of responses [53].

In addition, this study produced descriptions and expressions from participants [54], reflecting how they viewed the challenges and risks of the MSF ART programme. Attention was paid to unique themes that illustrated the range of the meanings of the phenomenon, in this case, the challenges in the implementation of the MSF ART programme, and how they could be addressed, rather than the statistical significance of the occurrence of particular texts or concepts [54]. The themes were obtained through triangulation of in-depth data and document review of sources of information by examining evidence from the authorities and using it to build a coherent justification for themes [51]. The data were coded using ATLAS.ti version 7.1.4, organising the data by highlighting texts, phrases, and words, and writing the word(s) or phrases representing a category in the margin [55]. In particular, the codes were developed based on emerging information generated from the participants [51].

The use of ATLAS.ti made the volume of work easier to manage, sort, and organise, store, annotate, and retrieve text, locate words, phrases, and segments of data, prepare codes networks, and extract quotes [56]. The software facilitated a systematic and rigorous data analysis process in comparing the data and triangulating different sources of data. In the process, it produced various networks among other codes, though the duty of explicating the data remained the researchers’ sole mandate. ATLAS.ti enhanced the triangulation of different data sources because we loaded all the transcribed interviews with secondary data as primary documents under the same hermeneutics and analysed them.

Various themes emerged and were further categorised under four major themes. The themes became central organising constructs, linked to a variety of related sub-themes. These themes were further explained and assigned meanings using thick description of the data [49,51].

## 3. Results

The participants indicated that the programme had challenges during and after the MoHCC takeover of the MSF ART programme. Despite these challenges, evidence from other studies reported that MSF ART provided high-quality services and established community participation. The main difficulties were parallel structures, lack of personnel resources, shortages of drugs, and donor dependency syndrome, as illustrated in Figure 1. It is imperative to indicate that this section only describes the results without much interpretation, the results are given meaning and significance under the subsequent Section 4.

### 3.1. Parallel Structures

The MSF programme brought in some remarkable changes to ART in both the Buhera and Tsholotsho districts. However, participants indicated that during its operations, it also faced some challenges, such as running parallel health structures and discord in policy decisions. For instance, some participants said MSF paid its workers in US dollars while the MoHCC paid its workers in Zimbabwean dollars, yet these workers were housed under one programme assigned to the same hospital. This parallel payment structure created tension between MoHCC and MSF staff, as articulated by one of the senior nurses:


*For your own information, the MSF staff were paid hefty salaries in US dollars, and yet we had the same qualifications. Those who were paid in Zimbabwean dollars were wallowing in poverty and were looked down upon by the MSF staff.*


Participants also highlighted that due to its lucrative payment structure, MSF attracted highly experienced nurses and doctors from the MoHCC. The health workers shunned the public health sector in favour of the MSF, which had a better remuneration structure.

In addition, participants indicated that MSF also provided ART beneficiaries with handouts, such as transport vouchers, for those who stayed far away from the hospital. This was meant to encourage them not to default due to transport challenges. MSF further referred beneficiaries to other NGOs who provided food in the area. The participants explained that this method ensured that the beneficiaries were well-fed, thus registering some weight gain. However, the incentives had a negative impact on other programmes as mentioned by one nurse in Tsholotsho:


*There were no gaps in the ART programme as such, but this programme superseded all other health programmes in the district because MSF supported it with incentives. This disadvantaged the performance of the district in programmes like EPI—there was under performance.*


Some of the participants indicated that MSF created policy discord instead of advocating and harmonising existing policies. For example, MSF created their own monitoring and evaluation systems which were not synchronised with MoHCC systems, as explained below:


*The MSF had a challenge of not following the ministry’s protocols and policies regarding monitoring and evaluation systems. For example, the organisation used Follow Up and Care of HIV infected and AIDS (FUCHIA) data collection and analysis system which contrasted with what the MoHCC used.*


Furthermore, participants stated that MSF came up with the policy changes without consulting the government, especially in relation to the threshold for ART initiation. For instance, the government policy stipulated that the threshold was CD4 cell count of ≤250 cells per microlitre and MSF changed it to a threshold which was CD4 cell count ≥250 cells per microlitre without consulting the government. This caused an influx of flow of HIV patients and strained the limited resources available in these two districts because the patients were put on ART before their CD4 count reached the government stipulated threshold. One participant explained:


*…they came with their own policies which sometimes went against the government policies with regards to ART initiation or had no or what can I say, the government had no idea of all those things. That was a challenge sometimes they came and say we initiate patients when CD4 is 1, 2, 3 whereas the government did not put into consideration that CD4 count threshold. Can that be made a policy and so forth? So, in terms of those things sometimes things were not flowing well according to the government policy.*


An interesting theme that emerged during the in-depth interviews was that there was a subtle but inherent conflict of interest, especially among the mentors from MSF who did not want to share their knowledge with those employed by the MoHCC. Indeed, the participants believed the MSF employees were not fully committed to sharing knowledge for fear of losing their jobs if they imparted the knowledge to non-MSF staff. Below is what was said by an MSF staff:


*As we were passing from a pure MSF supported approach to this mentoring of these nurses and capacity building it meant obviously that we are going to reduce the staff and that a number would lose their job. MSF staff would lose their job. So, I understood that in some instances, some staff employed by MSF would be kind of reluctant to transfer these competencies because it would mean now losing their job while MoHCC staff would now take up the knowledge and experience.*


### 3.2. Shortage of Manpower (Personnel Resources)

Besides parallel structures, in-depth interviews confirmed that the MSF ART programme also had personnel resource challenges. When the MSF ART programme started in Buhera and Tsholotsho districts, they brought their staff who were experts in handling the HIV and related Opportunistic Infections (OIs). Their staff comprised different experts ranging from doctors, nurses, logistic personnel, and other ancillary staff. Some were from outside the country whilst others were from within Zimbabwe. The provision of experts was timely as explained by key informants because they complemented the Ministry, which normally suffers from chronic shortage of medical personnel. In addition, it promoted exchange of skills and expertise among MSF and MoHCC staff as well as reducing the caseloads on the thin Ministry medical staff. However, when they left, they retrenched most of the staff, resulting in sustainability problems because there were increased volumes of work on the very few doctors and nurses who were available in MoHCC hospitals and clinics. One participant narrated:


*We used to have many nurses. So, MSF moved out and retrenched most of the nurses who used to help us in terms of labour here, so we are now understaffed, and we can feel the strain.*


Several participants further demonstrated how the sustainability of the programme was under threat because of the lack of personal resources to attend to HIV patients who were on ART. A nurse in Tsholotsho said:


*Now, there is a shortage of doctors because now there are only three doctors. We can have one doctor going for workshops, one going for male circumcision and only one doctor running the whole hospital, the Outpatient, the wards and the same doctor is the one running the district, she is the DMO (District Medical Officer) with that big responsibility on her shoulders, she is going to be in the office going for the meeting, this and that, it is so much and tiresome. So, the patients will be waiting only to be told at the end of the day that the doctor is no longer coming but with MSF the doctors had to takeover.*


Participants acknowledged that there were associated risks such as ART defaulting and loss of meagre resources by the patients as they waited for long hours to access ART. The patients receiving treatment were subjected to long hours of waiting for ART, while some were sent away without treatment at the end of the day.

The MSF staff had different perceptions to the MoHCC workers pertaining to retrenchment. To them, the retrenchments were necessary because they would facilitate distribution of more resources to the patients and cut on administrative costs. One of the participants working for MSF had this to say about retrenchments:


*We are doing this for the beneficiaries, because of the beneficiaries we are down scaling, that is happening now with MSF. What they are [MSF] doing is they are letting more and more people go, they are retrenching because they want to still maintain that status which I have told you that 85 to 90% must go to the beneficiaries ten to fifteen per cent then go to salaries and administration.*


### 3.3. Shortages of Drugs

The participants indicated that the programme’s sustainability after the withdrawal of aid by MSF was precarious since the government of Zimbabwe relied on donor funding for most of its health sectors to function. Therefore, most of the participants were afraid that there would be lack of financial sustainability since the country’s economy was ailing. For instance, OIs drug shortages were a common phenomenon in Buhera and Tsholotsho districts.


*Medication is no longer available. The main thing that we are doing is to say can you please go and buy this one, but by the time MSF was here most of the drugs were here. Now we are only running with few drugs. When MSF was still here, we were also going to get drugs, even for OIs, which were given for free.*


Another key informant concurred with the nurse in Buhera and highlighted those shortages of drugs manifesting soon after MSF withdrawal:


*About the sustainability of this programme, the major determining factor shall be finances to procure drugs. Given the financial situation in Zimbabwe it is going to be difficult to foresee this programme being sustainable, unless if they are going to bring another donor to help in that regard, I don’t know whether it will be sustainable now that they have withdrawn supplying the medication.*


Apart from shortages of drugs reported by the participants, there were also some critical activities funded by MSF that were no longer as functional as before. For instance, mobile clinics, which provided free HIV testing and education, had stopped functioning. This matter was raised by a senior nurse in Tsholotsho:


*MSF had some mobile programmes for HIV testing to give people free education on HIV/AIDS, but now we don’t see those mobile teams which were created by MSF again.*


### 3.4. Donor Dependency Syndrome

Despite the MSF having capacitated the MoHCC staff, the participants felt that the government and the two districts under study still did not have enough capacity to run the programme without the help of MSF donors. They still believed that MSF had all the necessary tools critical for the sustainability of the programme, such as finance, machines, and expertise. Some participants were of the view that the government should either allow MSF to continue or find another donor who would replace MSF. The following excerpts from the interviews conducted in both Buhera and Tsholotsho confirmed the dependency syndrome among the people and the dangers it posed to the sustainability of the programme. This was properly captured by the following interviews:


*I think the government should find another organisation or supportive partner who should be catering for some drugs or so on or MSF is supposed to come back in the district.*

*I can feel that we are going to have a problem of sustainability. The main obstacle, of course, continues to be money. The tests are expensive, the machines are expensive, and they require highly qualified staffs like the MSF who were here. On that note I would like to encourage the government to find another partner or bring MSF back.*


## 4. Discussion

This study elaborates the dynamic interface between governments and non-governmental organisations (NGOs). The results show that MSF ART programme faced some challenges during and after the programme was handed to the MoHCC. Among these challenges were parallel structures, personnel resources, shortages of drugs, donor syndrome, and conflicts of interest. These challenges are discussed within the context of the relationship between the NGOs and the government and corroborated by the literature.

### 4.1. Parallel Structures

It emerged that before MSF handed over the programme to the MoHCC it faced challenges in the running of parallel health structures. Some participants said MSF paid its workers in US dollars, while the MoHCC paid its workers in Zimbabwean dollars, yet these workers were all at the same hospital. This parallel payment structure created tension between MoHCC and MSF staff. Participants also highlighted that the lucrative payment structure by MSF attracted highly experienced nurses and doctors from the MoHCC. This is similar to the findings of [57,58,59] that donor initiatives, especially in the health sector, often cause internal brain drain of health workers from the public sector to better funded NGOs, consequently undermining the recipient’s institutional capacity. Furthermore, MSF also used to pay ART beneficiaries in the form of hand-outs such as transport vouchers for those who lived far away from the hospital. The government failed to meet this need and hence the challenge of defaulters, who did not adhere to treatment and missed collecting their medication.

Another example is monitoring and evaluation systems of the MSF, which were not synchronised with MoHCC systems. This result is fairly consistent with the findings of [60] in a study conducted in South Africa. They found that the services provided by donors were not well-integrated and, in many cases, created parallel structures for monitoring and evaluation.

The results also showed that MSF instituted policy changes without the knowledge of the government in relation to the threshold for ART initiation. For instance, the government policy stipulated that the threshold was ≤250 CD4 count, and MSF changed it to a higher threshold without consulting the government. This resulted in an influx of HIV patients in the Buhera and Tsholotsho districts because the patients were put on ART before their CD4 count reached the government stipulated threshold.

The issue of parallel structures created by donors is also confirmed by [31], who said that in Zimbabwe there is a problem of donors and other stakeholders setting up parallel structures. Ref. [31] further argued that the donors and other stakeholders in the health sector in Zimbabwe had been composed of different individual organisations, each with their own mandate and operational methods. As a result, they had not been cooperating with each other in a way which would allow coherent policy making [31]. As highlighted in this discussion, parallel structures by donors result in incoherent policy making and implementation, hence undermining the recipient’s institutional capacity.

### 4.2. Shortage of Personnel Resources

Besides parallel structural challenges, the results show that the MSF ART programme also had personnel resource challenges. When the MSF ART programme started in the Buhera and Tsholotsho districts, they brought their staff members who were experts in handling the HIV and related OIs. Their staff comprised different local and foreign experts ranging from doctors, nurses, logistic personnel, and other auxiliary staff. The provision of experts was prompt because they complimented the MoHCC, which normally suffers from a chronic shortage of medical personnel. In addition, it promoted the exchange of skills and expertise among MSF and MoHCC staff, and reduced the caseloads on the limited number of MoHCC medical staff. When the MSF left, the MoHCC retrenched most of the staff, which resulted in sustainability challenges. There were increased volumes of work that resulted in patients being occasionally turned down or subjected to long waiting hours, especially in Tsholotsho where MSF left earlier than Buhera. This risk had a ripple effect on the patients, such as defaulting and loss of meagre resources as they waited for long hours to access ART. This corroborates the findings by [61] who found human resource shortages, among others, as a challenge in health systems in Zimbabwe and Malawi. This is also confirmed by [62] who found that the level of staffing was extremely low in rural healthcare facilities of Zimbabwe, Mozambique, and Lesotho, which contributed to high proportions of patients per medical worker. The same can be said about South Africa, which has a severe shortage of healthcare personnel especially in the rural areas because of funding, historical deficiencies in infrastructure, and lack of opportunity, among other causes [63]. However, the MSF staff who participated in this study had different perceptions pertaining to the retrenchments. To them, the retrenchment was necessary because it would facilitate distribution of more resources to the patients and cut on administrative costs.

In a study conducted in Botswana about the transitional financing and the response to HIV/AIDS, the same challenge of lack of personnel emerged [64]. Furthermore, ref. [65] in a study in Windhoek verified that staff shortage was a hindrance to access of ART among HIV positive patients. This observation concurs with what the MoHCC of Zimbabwe cited as challenges in 2010, among them lack of financial, human, and material resources [43,66].

The challenge of long waiting hours to receive care and associated losses of income is consistent with results elsewhere in sub-Saharan Africa [60,67]. In addition, the result is fairly consistent with [68], the authors of which found that sub-Saharan Africa suffers from resource shortages leading to the overburdening of health systems and little time and resources to trace defaulters who could be brought back into the programme and avoid loss of life. Ref. [68] added that apart from resource inadequacy, sub-Saharan Africa also has a challenge of poor retention of skilled health workers, which is a big blow to the national treatment plan in most countries. According to [69], scarce financial and human resources and inadequate health-care infrastructure are stumbling blocks to the scaling up of treatment in sub-Saharan Africa. This study effectively highlights the shortage of personnel resources and implications on the quality of care provide to patients. The participants emphasised the emergency of longer waiting hours, increased defaulters, and loss of resources by patients as they wait to be treated.

### 4.3. Shortages of Drugs

The results of this study suggest that the MSF ART programme faced drug shortages; for instance, OIs drug shortages were a common phenomenon in Buhera and Tsholotsho districts soon after the donor handed over the programme to the MoHCC. The reason was poor financial support by the government, which was also experiencing massive economic challenges at the national level [43,66]. The findings also established poor procurement and distribution strategies soon after MSF left, especially in Tsholotsho district. Apart from shortages of drugs, there were also some critical activities funded by MSF which were no longer as functional as before. For example, mobile clinics which were used for free HIV testing and education had stopped functioning in the Tsholotsho district. The result complements the findings of [61] in research conducted in Zimbabwe and Malawi, which found severe drug shortages as one of the challenges. Studies elsewhere have also shown the same results; ref. [60] stated that resources are a major challenge in ART programmes. Their research in South Africa, Botswana, and Zimbabwe found that resources for pharmaceutical supply chains was a challenge, especially in Zimbabwe [60]. Hence, resources for drug procurement are limited and heavily affected by budgetary issues coupled with limited availability of foreign currency [60].

### 4.4. Donor Dependency Syndrome

Despite the MSF having capacitated the MoHCC staff, the results of this study indicate that participants reiterated that they still did not have enough capacity to run the programme without donors, in particular MSF. They still believed that MSF had all the necessary tools that were critical for the functionality of the programme such as finance, machines and expertise. Hence, some participants suggested that the government should either allow MSF to continue or find another donor who would replace MSF. This result confirms what many scholars have said about development aid, that it promotes dependency on others and creates the impression that eradication of poverty depends on the external donations rather than recipient’s own effort, motivation, arrangements, and institutions [19,70,71], as shown by the results of this study. Similar studies conducted on the effectiveness of development aid in HIV/AIDS indicate that donor dependency is particularly high for HIV programmes in low- and middle-income countries where HIV prevalence is high [72,73,74]; for example, combined donor assistance to HIV programmes in 127 countries amounted to 49% of total HIV funding in 2011 [72]. Donor dependency has long- term effects that may lead to a drastic reduction in quality and quantity of the treatment, care, and support for the ART patients. Therefore, the governments and NGOs need to collaborate to mitigate this dependency by putting in place dynamic and sustainable funding frameworks.

This study identified a variety of challenges calling for immediate resolution. Based on the results and discussion in the study, the following recommendations are proposed to make donor funded ART programmes more effective. These recommendations are also applicable in other areas. They help other government ministries and departments working directly or indirectly with donors, stakeholders such as local government, communities, and their leaders, and further development actors and academic institutions.

The study findings confirmed that the MSF ART programmes faced challenges such as running parallel health structures and creating policy discord, which created tension between MSF and the MoHCC. This resulted from lack of proper consultation, cooperation and integration. There is, therefore, an urgent need for donor programmes such as the MSF ART programme to align and harmonise their structures and policies with the recipient countries. This can be achieved by engaging and consulting with the national health Ministry to integrate their respective systems to the prevailing context. In this case study, the MSF was supposed to work with the MoHCC of Zimbabwe at the national, provincial, and district level to avoid discord through parallel structures. In summary there is need for promotion of effective communication and cooperation among all the actors involved in health delivery, such as NGOs, the local community, and national and local governments [1].

Besides parallel structure challenges, the findings show that the MSF ART programme also had personnel resource challenges. This study recommends that the MSF ART programme should foster a partnership with the recipient government to ensure an adequate budget for the remuneration of health staff. MSF and the MoHCC should attract skilled health personnel to non-urban areas characterised by harsh conditions through increased remuneration and rural allowances. It would be appreciated if all local employees were paid on the same scale, whether in the project or government. Institutional challenges such as different payment structures could have been resolved by donor stakeholder involvement and using the existing government payment structures. Donors will, therefore, give incentives where necessary and justifiable; for example, hardship allowances for those working under very harsh conditions, especially in the rural areas.

The study findings indicated that the MSF ART programme created a conflict of interest among the MSF mentors mentoring the MoHCC staff. The participants explained that the conflicts emanated from the fear of the MSF staff losing their jobs. In the future, the donor should fund and work with the MoHCC workers in their capacity as government staff and avoid conflicts of interest by not employing nurses from the Ministry of Health.

The results of this study point to a lack of financial sustainability, mainly to fund the procurement of much-needed drugs. Participants also expressed the need for MSF to continue funding the ART programme or to hand it over to another donor. This study recommends that, in the future, ART programmes should engage robust designs that support and develop the capacity and quality of drug suppliers in the recipient country. For example, NatPharm, in the case of Zimbabwe, should increase the availability of affordable drugs. This study further recommends that to sustainably capacitate the local pharmaceutical industry, donors should exploit existing potential and local resources to maximise cost-efficiency and increase self-reliance. In other words, donors should use local resources, which are cheaper in procurement and transport costs.

In addition, the ART programme should seek partnerships with economic programmes that have expertise in livelihood support and social protection initiatives [75] because the sustainability of the ART programme depends not only on the availability and provision of medicine but also on socio-economic support that guarantees social safety nets for those in need of support. As pointed out in this study, there are challenges, among them shortages of drugs for people who are on the ART programme and are trying to live with HIV as a manageable chronic condition. Some of these challenges can be solved if the programme seeks convergence with economic programmes.

Despite the support provided by the MSF ART programme, the results show that the MoHCC staff and community still needed more capacity to run the programme with the help of a donor, particularly the MSF. They still believed that the MSF had all the necessary tools for the ART programme, such as finance, machines, and expertise. Therefore, this study recommends that donors invest in strengthening the infrastructure of the health system that simultaneously addresses the major sustainability issues of the ART programme, such as ownership and participation by the MoHCC and the communities. More effort must be put into building the capacities of communities and creating effective local leadership that will ensure the sustainability of programmes when donors withdraw funding. Furthermore, the strategies should instil a sense of ownership at both national and local levels so that the programmes will still be sustained when donor funding ends.

When aid flows are volatile or unpredictable, programme plans are disrupted. The MoHCC and MSF should stimulate dynamic and sustainable funding systems for ART programmes; for example, the domestic revenue mobilisation recommended by the Organisation for Economic Cooperation and Development (OECDA) for fragile states [20]. Domestic revenue mobilisation is believed to be source of home-grown development finance [20], which could assist in predictable and sustainable funding of ART programmes. This can be enhanced by fostering seamless collaboration and interdependence between the public, private interests, and NGOs [1]. Since MSF does policy advocacy, it can also encourage the government of Zimbabwe to build accountable tax systems, especially for the administration and distribution of the AIDS levy, such that it will be able to fund ART programmes. For example, in 1994, the OECD/DAC agreed on ‘new orientations for development assistance’, emphasising the need for local control and long-term capacity development, followed by a call for a new partnership to reshape the 21st century. More recently, the World Bank (WB) and the International Monetary Fund (IMF) have moved from top-down structural adjustment programmes to a more participative process that brings local stakeholders together to help define national social and economic policies for poverty reduction. The resulting Poverty Strategy Papers (PRPs) have been used as the basis for decisions on aid and debt relief [76].

## 5. Conclusions

The case study explored the nuanced dynamics that govern the interface between the Zimbabwean state and MSF in the delivery of HIV/AIDS healthcare services, with a specific focus on the ART programme. The differences in terms of policies, structures, and administrative issues between the partners triggered challenges in the implementation of the programme. It shed light on the challenges that have arisen in the implementation of the MSF ART programme and the policy implications from the experiences of the programme. Some of the challenges identified were parallel structures, lack of personnel, shortages of drugs, donor dependency syndrome, and conflict of interests. This study has shown that each of these challenges discussed leads to disruptions in service delivery. This can increase the risk of defaulters, drug resistance, and deaths. Lack of responsiveness to the challenges may stifle ART programme sustainability and risk reversal of the gains attained so far in preventing, caring for, and treating HIV/AIDS. Therefore, more effort must be directed by the donor and government towards building the capacities of communities and creating effective local leadership that will ensure the sustainability of programmes when the donors withdraw their funding. These strategies should instil a sense of ownership at both national and local levels to continue providing the community with the required delivery of HIV/AIDS healthcare services. Above all, it is the government’s prerogative to supply public goods; therefore, the government of Zimbabwe must embark on a massive domestic revenue mobilisation to sustainably fund ART and other healthcare programmes.

This study has some limitations that provide the possibility for further research. The sample was mainly collected through convenience and purposive sampling techniques, which offers room for future research. Longitudinal studies might be needed to make causal inferences; for example, between shortages of drugs and the number of defaulters or lack of personnel and longer waiting hours before treatment. Furthermore, the findings of this study may not to be generalised across Zimbabwe because the setting in this study is predominantly rural. Therefore, there is need for a comparative study that will compare the urban areas and rural areas in Zimbabwe so that the results may apply to the generality of Zimbabwe.

## Figures and Tables

**Figure 1 ijerph-20-07137-f001:**
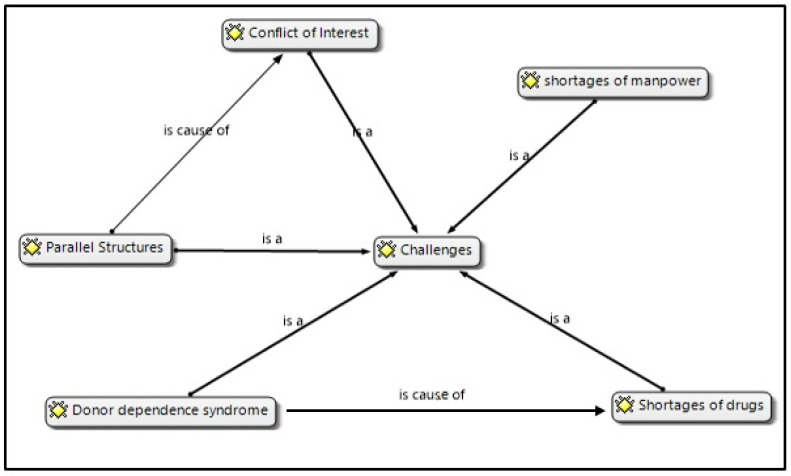
Network presentation of the results.

## Data Availability

The data presented in this study are available on request from the corresponding author. The data are not publicly available due to ethical reasons.

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
