# Peer review of "The Interface between the State and NGOs in Delivering Health Services in Zimbabwe—A Case of the MSF ART Programme"

_ijerph, 2023, doi:10.3390/ijerph20237137_

Round 1

Reviewer 1 Report

Comments and Suggestions for Authors

Dear Authors,

I have reviewed the manuscript entitled "The state-NGO interface in health service delivery in Zimbabwe. A case of the MSF ART programme". The manuscript is relevant to the nternational Journal of Environmental Research and Public Health, but needs to be revised before being considered for publication. I suggest making these changes:

Introduccion

La introducción es muy completa, sin embargo, se sugiere incluir estudios que permitan contextualizar más allá de China, asimismo, describir los principales factores de riesgo ambiental.

Materials and methods

The methodology is described in a broad and clear manner, which allows to have the elements to be replicated, however, the following is suggested:

- In the design of the study, it is suggested that a diagram be incorporated to make the procedure clearer so as to show each of the phases and what was carried out in them.

- Incorporate the inclusion and exclusion criteria.

- Incorporate information on informed consent.

Results

The results are descriptive and the opinions of the participants are evident; however, it is suggested to show a network diagram that allows visualizing the articulation of parallel structures, shortage of human resources and medicines, and the donor dependency syndrome, as well as the predominance of the perceptions of the participants.

Discussion

The discussion is complete; however, it is suggested to structure the discussions around the objective of the study, which will allow clarity on other similar studies and their articulation with their results, as well as the relationship of the challenges studied with non-governmental organizations and the government. It is also suggested to contrast studies carried out in urban areas and not only in rural areas.

Conclusions

It is suggested to conclude in terms of the objectives and hypotheses stated in the study.

References

The references are pertinent. 

In the hope that suggestions will improve your manuscript, I remain attentive to any response from you.

Best regards.

Author Response

Reviewer comment: La introduction es muy completa, sin embargo, se sugiere incluir estudios que permitan contextualizar más allá de China, asimismo, describir los principales factores de riesgo ambiental.

The introduction is very complete, however, it is suggested to include studies that allow contextualizing beyond China, as well as describing the main environmental risk factors.

Response:

The introduction gives background to the research topic, and it does not focus on China. (The introduction explains the dangers involved if the capacity to receive lifesaving aid is not adequate, and confusing and unclear political conditions make it impossible to judge the effectiveness of the aid)

Materials and methods

Reviewer comment: The methodology is described in a broad and clear manner, which allows to have the elements to be replicated, however, the following is suggested: - In the design of the study, it is suggested that a diagram be incorporated to make the procedure clearer so as to show each of the phases and what was carried out in them. - Incorporate the inclusion and exclusion criteria. - Incorporate information on informed consent.

Response

In brief, the study included only people who had direct knowledge and experience of the MSF programme either by being beneficiaries of MSF (workers with more than five years and ART patients), working for ministries of health (Nurses and medical doctors) and Local government (District administrators) and other organisations which operated in the same districts with MSF.

The paragraph below was revised, and the inclusion criteria explained from lines 305- 309

Two districts (Buhera and Tsholotsho) were selected by random sampling from the six districts in which the MSF project was implemented. The two districts are predominantly rural, and both have very high HIV prevalence compared to other districts in their respective province49. The people barely meet their nutritional requirements due to irregular food supplies49. In addition, the healthcare systems in both districts have a long history of getting funding from donors50. Secondly, purposive sampling was used to recruit participants into the study. Participants likely to offer valuable insights into the study were selected51. In brief, purposive sampling occurred at more than one level. First, purposive sampling was employed to choose key informants. The key informants were those considered to have more direct, personal knowledge of the MSF ART programme and government policies, also referred to as data-rich sources52. Secondly, the research questions and objectives were used to purposively identify and determine the type of participants to be recruited for in-depth interviews. (The research questions required someone with expansive experience in the programme and this is how the key informants were identified- based on their knowledge and experience ). In brief, the study included only people who had direct knowledge of the MSF programme either by being beneficiaries (workers with more than five years’ experience and patients), working for ministries of health (Nurses and medical doctors) and Local government (District administrators) or other organisations which operated in the same districts with MSF.

Results

Reviewer comment: The results are descriptive and the opinions of the participants are evident; however, it is suggested to show a network diagram that allows visualizing the articulation of parallel structures, shortage of human resources and medicines, and the donor dependency syndrome, as well as the predominance of the perceptions of the participants.

This study used thematic analysis and thick description to provide detailed accounts of the phenomenon understudy. We have added this network, figure number 1).

Authors’ construct using Atlas.ti (2016)

Discussion

Reviewer comment: The discussion is complete; however, it is suggested to structure the discussions around the objective of the study, which will allow clarity on other similar studies and their articulation with their results, as well as the relationship of the challenges studied with non-governmental organizations and the government. It is also suggested to contrast studies carried out in urban areas and not only in rural areas (the focus of the study is rural areas that is why it).

The study used thematic analysis which means identifying emerging themes from the data. So, we think that the best way to structure the discussion is to structure it around the themes which are central organizing themes.

On the issue of contrasting our study with studies carried out in urban areas- we have highlighted in line (762- 766) as our acknowledgement that this is a gap we recommend for further research.

The results are compared in section 4 (Discussion) still following the same themes that emerged. The discussion is based on the themes that emerged.

Conclusions

 Reviewer comment: It is suggested to conclude in terms of the objectives and hypotheses stated in the study.

Response: The study concluded in terms of the objectives and the themes which emerged from the data.

Reviewer 2 Report

Comments and Suggestions for Authors

The topic examined and discussed in the paper is very relevant and often remains hidden. LICs' aid in critical areas causes them to become dependent on donors, and the role of the companies and enterprises behind the NGOs participating in the process is often not clarified. This is the situation in many Sub-Saharan countries, such as Zimbabwe, which is particularly affected by HIV/AIDS. This analysis tries to go around the antiretoviral drug support program of MSF, the prestigious humanitarian organization, which has been running for more than 2 decades, and draw conclusions by placing it in a wider context.

The draft's introduction, which is longer than usual, explains the dangers involved if the capacity to receive lifesaving aid is not adequate, and confusing and unclear political conditions make it impossible to judge the effectiveness of the aid.

The qualitative research consisted of interviews with 40 program participants with open-ended questions. It would have been appropriate to present these questions in detail and to analyze the answers question by question. The data evaluation technique is described by the authors, but it is not clear whether the interviewees included only the local representatives of the receiving party or the staff of the donor NGO. Were all respondents asked the same set of questions? What was the role of the so-called informants?

It is not clear what exactly the role of the Ministry of Health in Zimbabwe was during the previous and current phases of the program, and how this has changed. It seems that parallel structures are operating, the ministry wants to intervene more strongly in the implementation without being prepared for it, while MSF seems to be pushed out of operational actions. All this is supported by the texts quoted from the interviewees.

Getting rid of donor dependency is an important aspect if national organizations and specialists are available to replace the donor, but this is not the case in the given country. It would be interesting to know which multinational business companies MSF cooperates with in the ART project in Zimbabwe, and what interests are represented here.

The authors are well aware of the limitations of the study (see conclusions), and recommend further investigations. In my opinion, this material also needs to be improved in several places, but due to the gravity of the problem raised, it is worth publishing with some revisions and amendments as proposed.

Author Response

Responses to the review comments

Reviewer comment: The qualitative research consisted of interviews with 40 program participants with open-ended questions. It would have been appropriate to present these questions in detail and to analyze the answers question by question.

Response: As highlighted in the materials and methods section this study used thematic analysis which means after we loaded all the transcribed interviews with secondary data as primary documents under the same hermeneutics in the Atlas.ti software we then explored patterns across the data. We pored across the data set to identify, analyse and report repeated patterns (Braun & Clarke, 2006). Various themes emerged and were further categorised under four major themes. The themes became central organising themes, which were linked to a variety of related sub-themes. These themes were further explained and assigned meanings using thick description of data.

Reviewer comment: The data evaluation technique is described by the authors, but it is not clear whether the interviewees included only the local representatives of the receiving party or the staff of the donor NGO. Were all respondents asked the same set of questions?

Response: In this study respondents/participants were asked different questions for example Key informants who had intimate knowledge about the MSF programme were asked questions that beneficiaries could not answer such as questions on the resources brought into the programme by donor aid and some policies and operational issues. Although interviews did not record the participants’ life history, increased attention was paid to past experiences and changes in the programme as fieldwork proceeded. In detail, emphasis was put on the knowledge they acquired, their experiences, and their understanding developed throughout the life cycle of the programme from 2000 to 2015. (Found from lines 333- 339)

In brief, the study included only people who had direct knowledge and experience of the MSF programme either by being beneficiaries of MSF (workers with more than five years’ experience and ART patients), working for ministries of health (Nurses and medical doctors) and Local government (District administrators) and other organisations which operated in the same districts with MSF.

Reviewer comment: What was the role of the so-called informants?

Response: The key informants were those considered to have more direct, personal knowledge of the MSF ART programme and government policies, also referred to as data-rich sources52. (found from lines 298- 300) they could deal with questions regarding resources, policies, administration etc.

It is not clear what exactly the role of the Ministry of Health in Zimbabwe was during the previous and current phases of the program, and how this has changed.

Response: They worked in partnership with each other as highlighted from lines 278- 281. After the 2013 The Ministry of Health was given autonomy to fund and manage the program as explained from lines 288- 290. Then this study was conducted soon after the handover to explore the challenges which occurred during the implementation of the MSF ART program, and how they could be addressed.

Reviewer comment: It would be interesting to know which multinational business companies MSF cooperates with in the ART project in Zimbabwe, and what interests are represented here.

Response: It would be interesting but according to MSF they are funded by millions of individual private donors who also give them the liberty to distribute the resources according to the needs of patients and not by the demands of individual donors. This makes very difficult to get the data regarding the multinational companies they cooperate with in the ART project in Zimbabwe, and what interests are represented.

Reviewer 3 Report

Comments and Suggestions for Authors

This case study explores the challenges exit between governments and NGOs through a qualitative research of MSF ART programme. The challenges should be think highly and the suggestion of the study give us many inspirations. And the problems should be

1.In the result part, some irrelevant words or sentences could be reduced, such as “ I think I talked a lot” in line 473.
2.In conclusion section, the study summed up the challenges were parallel structures, personnel resources, shortages of drugs, donor syndrome, conflicts of interest and religion. But the result part didn’t mentioned religion, this should be added and gave explanations for readers. 

Author Response

Responses to the review comments

Reviewer comment: And the problems should be;

1.In the result part, some irrelevant words or sentences could be reduced, such as “I think I talked a lot” in line 473.

The above was addressed in lines 453, 482 and 508.

Reviewer comment: 2. In conclusion section, the study summed up the challenges were parallel structures, personnel resources, shortages of drugs, donor syndrome, conflicts of interest and religion. But the result part didn’t mention religion, this should be added and gave explanations for readers.

The religion part was removed after the rigorous analysis. We found that it was an outlier that did not have enough evidence to support it in relation to the main objective of the current study. Regrettably, when we presented the results, we did not remove it. Now we removed it see line 755.

Round 2

Reviewer 2 Report

Comments and Suggestions for Authors

Although the authors made several corrections in the text, the purpose, results and conclusions of the research are still not clear enough for me. I sense what is being said behind the lines, the conflicts between the characters, but for the uninitiated, the picture is still unclear. I will not make a proposal to publish or reject the draft, I will leave it to the editor

Author Response

Dear Reviewer

Kindly find attached manuscript which was edited by a certified English editor.

Regarding your concern with the clarity of purpose, results and conclusion of the study. The purpose of this study and its relevance are explained from lines 217 to 232 as quoted here "Aid has not met its expected goals in terms of healthcare delivery due to political and economic challenges. The gap in aid administration has had severe consequences such as the intensification of diseases, untimely deaths, and unequal access to healthcare services. The failure to meet their expected goals has further aggravated more ravaging healthcare problems. That is, despite government and NGOs interventions, the problem persists. This brings to the fore the relevance of this study, which examines how the interface between the state and NGOs could curb the inherent problems in delivering health services in Zimbabwe. Research in this direction contributes to knowledge on how mutual collaborations between state and NGOs could salvage most of the health challenges confronting people in Zimbabwe. This is because most studies that have examined this similar conundrum have merely focused on the role of state and NGOs on an individual level. Consequently, there is little evidence and scant research on how a mutual connection between the state and NGOs could enhance the distribution of aid to resolve healthcare challenges in Zimbabwe. It is therefore worrisome that extant studies that have investigated this subject hardly interrogated why healthcare aid fails to achieve positive healthcare outcomes in Zimbabwe".

With regards to results, they were thematically presented and thick description was used for trustworthiness. Evidence from line 362 to 525. The results were further discussed from lines 526 to 728.

This study concluded that the differences in terms of policies, structures and administrative issues between the partners triggered challenges in the implementation of the programme. we also presented those challenges (see lines 732-739).

Thank you very much for your insightful review of our work.